# Color Quest: An interactive tool for exploring color palettes and enhancing accessibility in data visualization

**Luca Nelli** [iD] *

School of Biodiversity, One Health and Comparative Medicine, University of Glasgow, Glasgow, United Kingdom

* luca.nelli@glasgow.ac.uk

## Abstract

Data visualization plays a vital role in modern scientific communication across diverse domains, shaping the understanding of complex information through color choices. However, the significance of color palette selection goes beyond aesthetics and scientific communication, encompassing accessibility for all, especially individuals with color vision deficiencies. To address this challenge, we introduce "Color Quest," an intuitive Shiny app that empowers users to explore color palettes for data visualization while considering inclusivity. The app allows users to visualize palettes across various types of plots and maps envisioning how they appear to individuals with color blindness. In addition, it enables users to visualize palettes on their own custom-uploaded images. This short communication presents the app's design, interactive interface, and transformative potential in enhancing data visualization practices. Developed using open-source standards, Color Quest aligns with accessibility discussions, offering a practical tool and platform for raising awareness about inclusive design. Its open-source nature fosters transparency, community collaboration, and long-term sustainability. Color Quest's practicality renders it indispensable for scientific domains, simplifying palette selection and promoting accessibility. Its impact extends beyond academia to diverse communication settings, harmonizing information dissemination, aesthetics and accessibility for more impactful scientific communication.

**Data Availability Statement:** The full code is available in the supplementary material, and the most updated version is freely available at the author's github repository at https://github.com/lucanelli/colorquest.

## Introduction

Data visualization stands as an indispensable medium for conveying intricate information, serving as a cornerstone of modern communication across diverse domains [1]. The selection of colors within plots, graphs and maps, once viewed primarily through an aesthetic lens, has taken on a new dimension—one that encapsulates the nuances of accessibility, ethical considerations, and universal comprehension. This significance gains further weight when considering the prevalence of color vision deficiencies that impact a substantial portion of the population [2, 3]. The complications arising from certain color selections, such as the rainbow or jet color map, can inadvertently misrepresent data, with those affected by color vision

**Funding:** The author received no specific funding for this work.

**Competing interests:** The author have declared that no competing interests exist.

deficiencies often bearing the unintended repercussions of such choices in scientific communication [4, 5], which, in turn, highlights the critical need for innovative solutions to bridge this comprehension gap.

Recently, Stoelzle and Stein [6] undertook a meticulous review of color usage across scientific papers. Their analysis revealed that approximately that approximately 47% of publications in Hydrology and Earth System Sciences (HESS) in 2020 used visualizations that were not scientifically correct, perceptually uniform, and challenging for readers with color vision deficiencies. This trend seems persistent, as almost half of the 800 papers surveyed from 2005 to 2020 across the journal exhibited issues with ambiguous or non-colorblind-friendly color maps. Intriguingly, these tendencies were not confined to one journal; a similar survey in renowned journals such as Nature Scientific Reports and Nature Communications indicated that a significant proportion of papers across diverse disciplines also fell into the same trap. The authors speculate that this prevalence could be linked to a rise in spatial analyses or cartographic maps in research, often leading to misleading visualizations.

Given this prevalent challenge, an ethical imperative therefore emerges, especially in light of color vision deficiencies affecting a significant portion of the population [2, 3]. Achieving a harmonious equilibrium between visual allure and inclusivity is the crux of this endeavor [7].

Inspired by the call for accessibility considerations in color choices (e.g. [7–11]), I introduce "Color Quest", an intuitive Shiny app designed to aid users in creating accessible and visually appealing color palettes for data visualization. Color Quest offers users the ability to envision their chosen color palettes from the perspective of individuals with color blindness, by simulating scatter plots, line plots, box plots, histograms, and heatmaps. In addition, it enables users to upload their existing images and plots, providing simulations of how these visuals would appear to individuals with color vision deficiencies.

By addressing issues related to color accessibility, misuse, and inclusivity, "Color Quest" contributes to improving the overall quality and impact of scientific communication through data visualization.

This short communication explores the app's architecture, its interactive interface, and its potential to reshape data visualization practices.

## Methods

The most recent version of Color Quest is accessible at http://boydorr.gla.ac.uk/lucanelli/colorquest/.

Color Quest is developed using the R programming language [12] and integrates the 'shiny' [13], 'colourpicker' [14], 'colorBlindness' [15], and 'ggplot2' [16] packages. Once launched, the app presents a user-friendly interface with a single window, featuring multiple tabs (Fig 1). Users can select up to four colors using the provided color pickers located on the left sidebar. Upon scrolling down, users will reveal hexadecimal codes corresponding to their chosen colors. The app facilitates exploration across diverse visualization options housed under tabs like "Scatter Plot," "Line Plot," "Box Plot," "Histogram," and "Heatmap," each not only showcasing chosen colors but also offering simulations for various types of color blindness, including deuteranopia, protanopia, and desaturation. Although many color figures align well with individuals who have color blindness, there exists a limitation in terms of compatibility with black-and-white printers [5]. In light of this consideration, Color Quest takes a practical step by including a black and white (desaturation) option, to ensure visuals remain comprehensible even in grayscale print, enhancing inclusivity and accessibility across various modes of presentation.

Notably, the "Heatmap" tab empowers users to explore data's visual representation via a continuous color scale, enriching the spectrum of visualization possibilities.

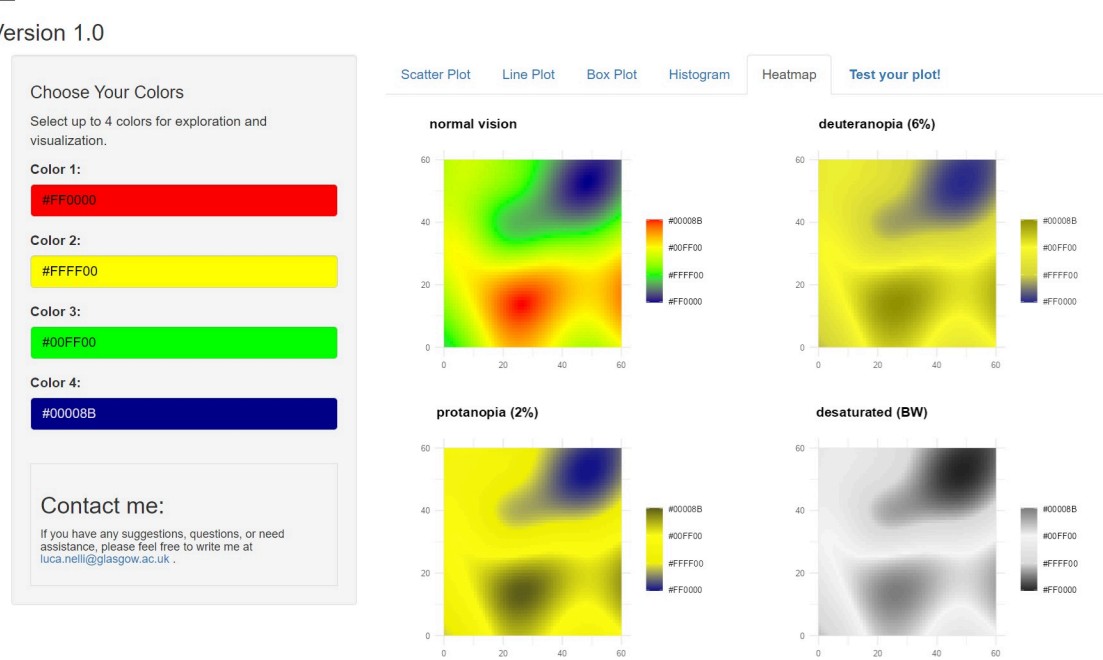

**Fig 1. Interface of the Color Quest app, freely available at http://boydorr.gla.ac.uk/lucanelli/colorquest/.**

An additional distinctive aspect of Color Quest lies in its ability to allow users to upload their own images (in jpg or png format), enabling users to evaluate how their plots, charts, diagrams, or other visual content, would appear to individuals with color vision impairments. Images are not saved on the server, ensuring users' security and privacy.

To ascertain the initial impact and usability of this tool, a user feedback mechanism was implemented within the app interface. This system captures live, anonymous user input concerning the tool's effectiveness and user-friendliness through a combination of quantitative ratings and qualitative comments.

In addition to desktop usage, Color Quest is optimized for seamless visualization on tablets and smartphones, enabling users to directly capture images of existing plots using their phones. This feature ensures accessibility and convenience across various devices.

Both the code and its documentation can be found in the supplementary material and are continuously updated on my GitHub repository: https://github.com/lucanelli/colorquest. On GitHub, users will find detailed documentation, step-by-step instructions, explanations of code functions, and use-case examples.

## Discussion

Color Quest aligns with ongoing discussions about accessibility in data visualization [7]. It serves as both a practical tool for data visualization enthusiasts and a platform for raising awareness about the significance of inclusive design. The app's versatility is showcased through simulations that illustrate how various visualization types, including scatter plots and histograms, would appear to individuals with color vision deficiencies and in black and white [5], fully answering the call for inclusivity in scientific communication [4].

The app's utility extends beyond academia to practical contexts like presentations, posters, conferences, and lectures, offering a broader impact in various communication settings. In

particular, the users' ability to upload their pre-existing plots and figures, gives Color Quest a versatility that spans a range of scientific domains. For instance, in medicine, Color Quest can be helpful in future genetic research studies, ensuring that subtle genetic variations visualized are discernible by researchers across the spectrum of color vision. In epidemiology, should a future outbreak arise where data visualization plays a critical role in decision-making, Color Quest could be instrumental in crafting universally comprehensible color-coded infection rate maps, facilitating more informed and rapid response strategies. Similarly, in ecology and biodiversity conservation Color Quest may play a transformative role by ensuring that maps detailing biodiversity hotspots or environmental degradation areas are accessible to policymakers and the general public, irrespective of their color vision capabilities.

The software's open-source nature, rooted in principles of transparency, community collaboration, and ongoing improvement, allows researchers, developers, and the community to both benefit from its features and contribute to its evolution, ensuring long-term sustainability, adaptability, and fostering a collaborative innovation ecosystem.

## Conclusions

Color Quest's development and deployment mark a significant step in data visualization. By harmonizing aesthetics with accessibility, the app empowers users to select colors that align with their narrative while ensuring inclusivity. The impact of this tool extends beyond boundaries, enabling equitable communication of insights to diverse audiences. In an increasingly data-centric world, the evolution of tools like Color Quest contributes to visually engaging and universally comprehensible information dissemination.

## Supporting information

**S1 File. Striking image.**
(PNG)

**S1 Text.**
(TXT)

## Acknowledgments

The 'Striking image' used to represent this article online and on the journal homepage was generated using advanced artificial intelligence (AI) technologies, specifically ChatGPT and DALL-E, developed by OpenAI (https://www.openai.com).

## Author Contributions

**Conceptualization:** Luca Nelli.

**Data curation:** Luca Nelli.

**Formal analysis:** Luca Nelli.

**Methodology:** Luca Nelli.

**Software:** Luca Nelli.

**Visualization:** Luca Nelli.

**Writing – original draft:** Luca Nelli.

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
