## [Decision Letter · Decision Letter 0]

18 Sep 2023

PONE-D-23-26146Color Quest: An Interactive Tool for Exploring Color Palettes and Enhancing Accessibility in Data VisualizationPLOS ONE

Dear Dr. Nelli,

Thank you for submitting your manuscript to PLOS ONE. After careful consideration, we feel that it has merit but does not fully meet PLOS ONE’s publication criteria as it currently stands. Therefore, we invite you to submit a revised version of the manuscript that addresses the points raised during the review process.

We look forward to receiving your revised manuscript.

Kind regards,

Mohamed Rafik N. Qureshi, Ph.D.

Academic Editor

PLOS ONE

2. Please remove your figures from within your manuscript file, leaving only the individual TIFF/EPS image files, uploaded separately. These will be automatically included in the reviewers’ PDF.

3. We note that Figure 1 in your submission contain copyrighted images. All PLOS content is published under the Creative Commons Attribution License (CC BY 4.0), which means that the manuscript, images, and Supporting Information files will be freely available online, and any third party is permitted to access, download, copy, distribute, and use these materials in any way, even commercially, with proper attribution. For more information, see our copyright guidelines: http://journals.plos.org/plosone/s/licenses-and-copyright.

A. You may seek permission from the original copyright holder of Figure 1 to publish the content specifically under the CC BY 4.0 license. 

B. If you are unable to obtain permission from the original copyright holder to publish these figures under the CC BY 4.0 license or if the copyright holder’s requirements are incompatible with the CC BY 4.0 license, please either i) remove the figure or ii) supply a replacement figure that complies with the CC BY 4.0 license. Please check copyright information on all replacement figures and update the figure caption with source information. If applicable, please specify in the figure caption text when a figure is similar but not identical to the original image and is therefore for illustrative purposes only.

Additional Editor Comments:

A significant improvement is necessary to the paper "Color Quest: An Interactive Tool for Exploring Color Palettes and Enhancing Accessibility in Data Visualization" since it fails to meet a number of quality criteria.

Reviewers' comments:

Reviewer's Responses to Questions

**Comments to the Author**

1. Is the manuscript technically sound, and do the data support the conclusions?

Reviewer #1: No

Reviewer #2: Yes

2. Has the statistical analysis been performed appropriately and rigorously? 

Reviewer #1: No

Reviewer #2: N/A

3. Have the authors made all data underlying the findings in their manuscript fully available?

Reviewer #1: No

Reviewer #2: Yes

4. Is the manuscript presented in an intelligible fashion and written in standard English?

Reviewer #1: Yes

Reviewer #2: Yes

5. Review Comments to the Author

Reviewer #1: Authors of this document asserted an Interactive Tool for Exploring Color Palettes and claiming that this can enhance Accessibility in Data Visualization.

This paper has a good structure with a good description about the proposed solution which looks promising but it doesn’t provide any examples of using it. References are from reputable publishers and also up-to-date.

No literature review was included, which was intended to offer a thorough grasp of the research problem and the various existing or similar solutions. This paper also lacks both metrics for assessing the proposed solution and a verified analysis of its efficacy.

Reviewer #2: The manuscript introduces "Color Quest", an interactive tool developed to address accessibility issues in data visualization, specifically considering individuals with color vision deficiencies. This is a significant contribution, given the growing emphasis on inclusivity and accessibility in scientific communication.Overall, the manuscript is well-structured and written clearly. The abstract succinctly summarizes the paper's main points and provides a clear understanding of the tool's purpose and potential impact.

Specific Comments:

Line 5: "Glsagow, UK" seems to be a typo. It should be corrected to "Glasgow, UK".

Lines 12-13: You mentioned the ability of the tool to visualize color palettes on custom images. Including a brief mention of the feature allowing users to upload their images might provide a more rounded view of the app’s capabilities in the abstract.

Line 66: The connection between the limitation with black-and-white printers and the black and white option in the tool could be clarified further.

Lines 78-79: Mentioning that the code is open-source and available at GitHub is a great point. It would be good to also state if any documentation or community guidelines are available for potential contributors.

Lines 80-95: The discussion highlights the potential impacts of the tool very well. Adding a glimpse of future developments or enhancements could add more depth to this section. E.g., mentioning a few specific examples or case studies where the tool could be beneficial might give a more grounded view of its potential applications.

6. PLOS authors have the option to publish the peer review history of their article (what does this mean?). If published, this will include your full peer review and any attached files.

Reviewer #1: No

Reviewer #2: No

---

## [Decision Letter · Decision Letter 1]

16 Nov 2023

Color Quest: An Interactive Tool for Exploring Color Palettes and Enhancing Accessibility in Data Visualization

PONE-D-23-26146R1

Dear Dr. Nelli,

We’re pleased to inform you that your manuscript has been judged scientifically suitable for publication and will be formally accepted for publication once it meets all outstanding technical requirements.

Kind regards,

Mohamed Rafik N. Qureshi, Ph.D.

Academic Editor

PLOS ONE

Additional Editor Comments (optional):

Thank you for your updated version of "Color Quest: An Interactive Tool for Exploring Color Palettes and Enhancing Accessibility in Data Visualization"

Reviewers' comments:

Reviewer's Responses to Questions

**Comments to the Author**

1. If the authors have adequately addressed your comments raised in a previous round of review and you feel that this manuscript is now acceptable for publication, you may indicate that here to bypass the “Comments to the Author” section, enter your conflict of interest statement in the “Confidential to Editor” section, and submit your "Accept" recommendation.

Reviewer #2: All comments have been addressed

Reviewer #3: All comments have been addressed

2. Is the manuscript technically sound, and do the data support the conclusions?

Reviewer #2: Yes

Reviewer #3: Yes

3. Has the statistical analysis been performed appropriately and rigorously? 

Reviewer #2: N/A

Reviewer #3: N/A

4. Have the authors made all data underlying the findings in their manuscript fully available?

Reviewer #2: Yes

Reviewer #3: Yes

5. Is the manuscript presented in an intelligible fashion and written in standard English?

Reviewer #2: Yes

Reviewer #3: Yes

6. Review Comments to the Author

Reviewer #2: The revision is sufficient. Further information on the related literature has been provided, future developments have been addided to the Discussion and grammatical issues have been corrected.

Reviewer #3: I thank the author for their effort and time in revising this paper. Please consider fixing this minor typo in the manuscript:

"that approximately that approximately" → "that approximately"

7. PLOS authors have the option to publish the peer review history of their article (what does this mean?). If published, this will include your full peer review and any attached files.

Reviewer #2: No

Reviewer #3: **Yes: **Ghazal Kalhor

---

## [Editor Report · Acceptance letter]

9 Jan 2024

PONE-D-23-26146R1 

PLOS ONE

Dear Dr. Nelli, 

I'm pleased to inform you that your manuscript has been deemed suitable for publication in PLOS ONE. Congratulations! Your manuscript is now being handed over to our production team.

Kind regards, 

on behalf of

Prof.(Dr.) Mohamed Rafik N. Qureshi 

Academic Editor

PLOS ONE